# The Role of Serotonin Neurotransmission in Gastrointestinal Tract and Pharmacotherapy

**DOI:** 10.3390/molecules27051680

**Published:** 2022-03-03

**Authors:** Tomasz Guzel, Dagmara Mirowska-Guzel

**Affiliations:** 1Department of General, Gastroenterology and Oncologic Surgery, Medical University of Warsaw, Banacha 1a, 02-097 Warsaw, Poland; tomasz.guzel@wum.edu.pl; 2Department of Experimental and Clinical Pharmacology, Medical University of Warsaw, Banacha 1b, 02-097 Warsaw, Poland

**Keywords:** serotonin, 5-HT receptors, gastrointestinal tract

## Abstract

5-Hydroxytryptamine (5-HT, serotonin) is a neurotransmitter in both the central nervous system and peripheral structures, acting also as a hormone in platelets. Although its concentration in the gut covers >90% of all organism resources, serotonin is mainly known as a neurotransmitter that takes part in the pathology of mental diseases. Serotonin modulates not only CNS neurons, but also pain transmission and platelet aggregation. In the periphery, 5-HT influences muscle motility in the gut, bronchi, uterus, and vessels directly and through neurons. Serotonin synthesis starts from hydroxylation of orally delivered tryptophan, followed by decarboxylation. Serotonin acts via numerous types of receptors and clinically plays a role in several neural, mental, and other chronic disorders, such as migraine, carcinoid syndrome, and some dysfunctions of the alimentary system. 5-HT acts as a paracrine hormone and growth factor. 5-HT receptors in both the brain and gut are targets for drugs modifying serotonin neurotransmission. The aim of the present article is to review the 5-HT receptors in the gastrointestinal (GI) tract to determine the role of serotonin in GI physiology and pathology, including known GI diseases and the role of serotonin in GI pharmacotherapy.

## 1. Introduction

Serotonin (5-hydroxytryptamine; 5-HT) and serotoninergic drugs have become popular in the treatment of depression, and this common use has contributed to a broad knowledge of serotoninergic neurotransmission in the central nervous system (CNS). However, serotonin is not found solely in the CNS, and its highest concentrations occur elsewhere. Though obtained from a normal diet, serotonin must be synthesized de novo from tryptophan at the destinations of its action because it is almost fully metabolized before being absorbed. Tryptophan (TRP) is a nutritional amino acid and precursor of many physiologically essential substances, including 5-HT, melatonin, and kynurenine. Only 1–2% of TRP is degraded into 5-HT and melatonin, as 95% goes into kynurenine, kynurenic acid, xanturenic acid, quinolinic acid, and picolinic acid through the kynurenine pathway [1]. TRP undergoes hydroxylation in neurons and chromaffin cells via the enzyme tryptophan hydroxylase (TPH), which exists as two isoforms: TPH1 localized in the gut and TPH2 found in CNS neurons, mainly the raphe nuclei of the brain stem, and some neurons in the enteric nervous system (ENS). 5-Hydroxytryptophan (5-HTP) is then decarboxylated by a non-specific aromatic L-amino acid decarboxylase (L-AADC) that also participates in histamine and catecholamine transformation. 5-HT compounded with chromogranin A (CGA) is stored in vesicular monoamine transporter 1 (VMAT1), which is expressed by granules/vesicles in enterochromaffin (EC) cells [2]. EC cells within the gastrointestinal (GI) mucosa synthesize and secrete up to 95% of total body serotonin, whereas the remaining 5% is synthesized by neurons, mostly in the CNS, but also in pancreatic islets, mammary glands, and adipose tissue [3]. These cells respond to chemical and mechanical stimulation, but also collect signals from gut microbiota to release serotonin [4]. 5-HT interacts with nerve terminals or other cells (immune, epithelial) or is transported into enterocytes by the serotonin reuptake transporter (SERT) or organic cation transporters and dopamine transporter when SERT does not work. Elimination occurs through deamination to 5-hydroxyindole acetaldehyde (5-HIAL), followed by oxidation to 5-hydroxyindoloacetic acid (5-HIAA), which is excreted by the kidneys. The urine concentration of 5-HIAA is clinically important as an indicator of 5-HT synthesis [5]. A scheme of serotonin production is presented on Figure 1 [6].

Serotonin was first discovered in the bowel and called “enteramine” by Vittorio Espramer in 1937 [7]. Further studies showed that “enteramine” has the same structure as the serum vasoconstrictor described by Rupport in 1947, which was called “serotonin” [8]. Serotonin and its receptors are part of the multimodal processes that affect homeostasis. 5-HT receptors are classified into seven families involving at least 15 subgroups of receptors [9]. Only the 5-HT3 receptor is a cation channel activated by a ligand; the others are coupled to a G protein, which is a membrane protein consisting of three subunits (alpha, beta, gamma), interacts with guanine nucleotides (GTP and GDP), and conjugates with different receptors that may result in several intracellular effects. The most common type of receptors in the GI tract are 5-HT3, 5-HT4, and 5-HT7. 5-HT receptors and their functions are presented in Table 1.

Serotonin increases the motility of the GI tract muscles, induces muscle constriction in the lungs and uterus, influences vessel muscles in both directions (constriction/relaxation), takes part in platelet aggregation, excites nociceptive pain neurons, and influences CNS neurons. Serotonin also plays a role in the symptoms of GI inflammation, acting through different mechanisms to exert pro- or anti-inflammatory activity [5,10,11,12]. Common substances have been shown to influence serotonin secretion, which may be responsible for the presence of ambiguous disease symptoms and prevalent digestive diseases. Bisphenol A, a chemical compound in plastic bottles made of PE, was shown in an animal study to enhance the number of 5-HT-positive cells in the mucosal layer of the small intestine [13]. GI tract disorders are closely associated with serotoninergic transmission. We review the current knowledge of peripheral serotonin neurotransmission with special emphasis on the effects essential for GI action and reference to drugs affecting the alimentary system.

## 2. Serotonin in the GI Tract

A highly organized ENS, also called the “abdominal brain” or “second brain”, regulates absorption, secretion, and gut motility. It is the only organ that, although under the control of the CNS, can function in isolation independent of the brain. Serotonin is secreted by EC cells apically to the intestine lumen and baso-laterally in response to mechanical and chemical stimulation [14,15]. Through the mucosal projections of primary afferent neurons, intrinsic and extrinsic nerves conduct information to the CNS. Submucosal projections of intrinsic primary afferent neurons (IPANs) initiate peristaltic and secretory reflexes, whereas myenteric projections are responsible for giant contractions and mediate excitatory neurotransmission regulating GI motility [16]. Mechanical stimulation of the bowel (by food) provides secretion of 5-HT that activates IPANs containing calcitonin gene-related peptide (CGRP) in the mucosa, which synapse with ascending and descending interneurons. Through excitatory motor neurons, ascending interneurons cause contraction of smooth muscle mediated by acetylcholine, substance P, and neurokinin A. Descending interneurons cause smooth muscle relaxation, activating inhibitory motor neurons to release nitric oxide (NO), vasoactive intestinal peptide (VIP), and pituitary adenylate cyclase-activating peptide (PACAP) [17]. Longitudinal smooth muscles contract and relax in reverse fashion to circular muscle under the regulation of the same neurotransmitters [18,19].

### 2.1. GI Serotonin—Linkage to Metabolism

Investigations on human carcinoid EC cell lines have given information about its response to some nutrients. It was confirmed that EC cells are sensitive to luminal glucose levels through glucose–sensing mechanisms for both an acute increase and chronic reduction in glucose availability. EC cells express a range of G-protein-coupled receptors that are tuned to other nutrients, such as amino acids triggering serotonin release. Similarly, production of secondary bile acids and short-chain fatty acids by gut microbiota activates EC cells for 5-HT secretion. In vitro EC cells respond to neuromodulatory agents, which suggests that these cells may be responsive to neural signals in the CNS and ENS. Luminal 5-HT is required for intestinal nutrients and water absorption and takes part in bicarbonate and electrolyte secretion into the lumen, as well as bile acid turnover, bile acid synthesis, and liver secretion [20].

Serotoninergic intestinal activity depends on local stimulation. 5-HT secreted by EC cells acts as a paracrine messenger because there are no direct connections to the activated nerves. There is also confirmed hormonal signaling with other enteroendocrine cells. Mouse models have shown the role of gut-derived serotonin (GDS) in lipolysis and gluconeogenesis. Sumara et al., identified that 5-HT acts directly on adipocytes through the Htr2b receptor and promotes lipolysis during fasting. As a result, free fatty acids and glycerol are released as a substrate for liver gluconeogenesis. Htr2b is the most highly expressed 5-HT liver receptor. Serotonin has been observed to favor gluconeogenesis by enhancing the activity of two rate-limiting enzymes, FBPase and G6Pase. Through the same receptor, GDS suppresses glucose uptake in hepatocytes, possibly by suppressing glucokinase activity and glucose transporter 2 (Glut2) degradation. In diabetes type 2, there is often lipolysis and liver gluconeogenesis, and these results suggest that decreasing GDS synthesis could ameliorate type 2 diabetes [21]. Some EC cells express specific receptors, including receptors for glucagon-like peptide 1 and 2 (GLP1, GLP2) in diabetes and GI tract motility, respectively. GLP2 is secreted in response to food intake and influences GLP1 secretion from L cells in the ileum in response to an elevated glucose concentration. Thus, serotonin may act as a kind of mediator in inhibiting gastric emptying and stimulating gastric secretion with GLP2, and takes part in the metabolic reaction for glucose intake with GLP1 [20]. The intestinal microbiome performs a range of essential functions on the host metabolism.

Although the gut bacterial system, including that of Corynebacterium spp., Streptococcus spp., and Enterococcus spp., can synthesize and secrete 5-HT, it provides signals through metabolites, such as fatty and biliary acids, to EC cells to maintain the gut content and plasma serotonin levels [22].

Only a small amount of 5-HT is synthesized with bacterial beta-glucuronidase from glucuronide-conjugated serotonin. Some microbial metabolites promote TPH1 expression and 5-HT release. The pathway goes through specific receptors present in the colon, including olfactory receptor 78 (OLF78) for short-chain fatty acids, G-protein-coupled receptor 35 (GPR35) for small aromatic acids, G-protein-coupled bile acid receptor 1 (GPBAR1) for secondary bile acids, free fatty acid receptor 2 (FFAR2), and other G-protein coupled receptors [23,24]. The enteric microbiota is crucial and takes part in regulating a variety of neurotransmitters, including those playing an important role in diseases of the GI tract, such as histamine, serotonin, and glutamate. Non-specific and specific symptoms of GI tract failure seem to be a common problem in health services all over the world. According to the National Centre for Health Statistics (NCHS) in the US, there were 37.2 million visits to physician offices and 7.9 million to the emergency department with diseases of the digestive system as a primary diagnosis in 2018 [25].

### 2.2. Serotonin in GI Tract Disorders

#### 2.2.1. Irritable Bowel Syndrome (IBS)

IBS is a bothersome disease that may occur with several leading symptoms with predominant complaints according to Rome IV criteria of constipation (IBS-C), diarrhea (IBS-D), mixed type (IBS-M), and unclassified (IBS-U) [26]. Symptoms may vary and are dependent on diet and lifestyle, and even may be influenced by psychological factors. There are no reliable, established, standardized tests to assess disease activity. One of the crucial factors that may influence the disease course is intestinal microbiota, and communication between the gut microbiota and CNS was proposed as the brain-gut microbiome (BGM). The microorganism composition is relatively stable, and an imbalance in homeostasis caused by usage of antibiotics or other drugs, diet change, and/or exposure to enteritis may lead to dysbiosis and clinical deterioration [27,28]. Bacterial overgrowth in the small intestine (SIBO) leads to a disturbance in intestinal absorption and release of toxic products, such as methane gas, influencing motor activity and delayed bowel transit [22]. There are several microbial neurotransmitters, including 5-HT, which seems to be an essential molecule that induces IBS symptoms through the activation of immune cells, changes in gut motility, and increased visceral hypersensitivity and epithelial permeability. Excessive serotonin release promotes GI motility and may be responsible for IBS-D symptoms, and bacteria-derived 5-HIAA directly accelerates colonic motility via activation of L-type calcium channels [29]. In contrast, increased SERT function and downregulation of TPH1 in animal models has been shown to reduce GI motility and lead to delayed intestinal transit [30,31]. Through 5-HT as a neurotransmitter, the gut microbiota stimulates mesenteric sensory fibers and vagal and spinal afferent fibers, influencing the presence of chronic pain in IBS patients, and the pro-inflammatory action of serotonin exacerbates abdominal pain intensity [32,33].

#### 2.2.2. Inflammatory Bowel Disease (IBD)

IBD consists of two diseases—ulcerative colitis (UC) and Crohn’s disease (CD). CD may affect all of the small and large bowel, whereas UC is limited to the colon. There are different types of immune responses involved in GI inflammation. CD is characterized by type 1 or type 17, with increased production of IL-12, IL-17, IL-23, IFNγ and TGFβ, whereas UC is typified by type 2 with higher levels of IL-5 and IL-13 [34,35]. There is high symptom heterogeneity, and the main factors affecting the course of IBD are genetic predisposition, infectious history, diet, and microbiota, which seems to play an important role in intestinal inflammation [36]. Kwon et al., compared the influence of serotonin on chemically induced colitis in a mouse model and showed that serotonin selected for more colitogenic microbiota by regulating bacterial growth and inhibiting beta-defensin production in colonic epithelial cells. A significantly reduced intestinal concentration of 5-HT decreased the severity of colitis. Microbial transfer between guts with 5-HT reduced and normalized the mediated colitis severity, indicating the protective function of microflora with reduced serotonin levels [37]. In humans, *Escherichia coli*, available as probiotic bacteria, regulates tryptophan hydroxylase 1, enhancing the 5-HT level. It may be beneficial in IBS-C to have a decreased serotonin level, and in IBD, an increased level of serotonin may be associated with exacerbation of the clinical course due to enhanced inflammation [38]. Microbiota-derived short-chain fatty acids, through upregulation of G-protein coupled receptors, strengthen the epithelial barrier integrity and, by inducing regulatory T cells, have anti-inflammatory potential. On the other hand, short-chain fatty acids upregulate TPH transcription, which increases mucosal production of serotonin, supporting inflammation and colitis manifestation [26].

Peripheral serotonin is important for a proper immune response but also impacts various inflammatory conditions, such as IBD. The main source of 5-HT for the immune response is platelets. Platelets cannot produce 5-HT themselves, but they take it from the bloodstream by SERT. In contrast, mast cells, monocytes/macrophages, and T cells partially participate in serotonin production. Serotonin acts as a chemotactic agent, increases pro-inflammatory cytokine secretion (e.g., IL-1, IL-6, IL-8, and NFkB), and enhances phagocytosis. Serotonin can affect both T cells and lymphocytes through their 5-HT receptors [39,40,41,42]. 5-HT increases the production of reactive species, enhances cytokine production, and promotes adhesion of monocytes to GI epithelial cells [43].

#### 2.2.3. Carcinoid

Carcinoid is one of the best described neuroendocrine tumors (NETs), and carcinoid syndrome (CS) is the most frequent of the NET ectopic hormonal syndromes. CS occurs when a sufficient amount of tumor-related compounds are secreted into systemic circulation. Despite the fact that numerous potential mediators are reported to be responsible for the clinical symptoms of CS, the most frequently used for laboratory confirmation is 5-HIAA, the product of 5-HT degradation. Overproduction of serotonin is noted in 98–100% of cases, and persevering diarrhea is one of the symptoms of CS [44]. Serotonin stimulates colonic motor functions via 5-HT3 and chloride ion secretion via the 5-HT2A and 5-HT4 receptors [9].

In February 2017, the Food and Drug Administration (FDA) approved telotristat ethyl for the treatment of CS diarrhea in patients inadequately controlled by somatostatin analogue therapy. The European Medicine Agency (EMA) followed suit in September of that year. The drug is a first-in-class, small molecule, peripheral TPH inhibitor [45]. TPH is one of the main causes of the effects of CS, such as frequent bowel movements associated with diarrhea and other symptoms [46]. Telotristat ethyl is given as a tablet three times daily and is generally well tolerated. The proportion of patients reporting at least one adverse event related to the drug is similar to that observed with placebo. The most common adverse events are abdominal pain, nausea, increased gamma-glutamyltransferase, and fatigue [47]. However, the drug was labeled with a black triangle due to EMA regulations for new drugs and biologics approved since January 2011. The symbol displayed in their package leaflet and the summary of product characteristics (SmPC) for healthcare professionals means that the medications are under ‘additional monitoring’ by regulatory authorities, and is meant to encourage doctors and patients to report any adverse effects [48].

### 2.3. 5-HT Receptors and Drugs Acting in the GI Tract

Multimodal serotonin GI function and enteric pathologies are a result of the presence of different types of 5-HT receptor families. 5-HT1 includes five subtypes: 5-HT1A, 5-HT1B, 5-HT1D, 5-HT1e, and 5-HT1F. However, for 5-HT1e, a functional response in native cells or tissue has not been identified [9]. What was the 5-HT1C receptor was finally classified as 5-HT2C, so it does not exist in the 5-HT1 family. Representatives of this group of receptors have been confirmed in many tissues. The most extensively distributed is the 5-HT1A receptor, which is present in the CNS, cardiovascular system, and GI tract. Extensive investigations of the distribution of the 5-HT1 receptors and their involvement in a variety of physiological and pathological responses have led to a broad knowledge base on 5-HT1 receptor identification, expression, and pharmacology. Much is known about 5-HT1A receptors, especially in area of targeting for pharmacotherapy in a broad spectrum of neuropsychiatric disorders, such as major depression, anxiety, schizophrenia, pain, attention deficiency hyperactivity disorder, cognitive deficits, Parkinson’s disease, and recently sexual dysfunction and respiratory deficits [9]. Its role in the human GI tract seems to be rather limited. The main confirmed role is a contribution to gastric contractility. Application of a selective 5-HT1A receptor agonist, R137696, resulted in fundus relaxation with no effect on distension-induced dyspeptic symptoms [49]. Similar observations concerning the administration of sumatriptan, a 5-HT1D agonist, was reported by Coullie et al., who noted that relaxation of the gastric fundus resulted in a prolonged half-emptying time of liquids and solids [50]. Sumatriptan also increases lower esophageal sphincter (LES) contraction but increases the frequency of reflux. This is probably due to sumatriptan evoking stomach postprandial relaxation and delayed meal retention [51]. The 5-HT1B receptor is present in the CNS and vessels, but 5-HT1B/1D agonists have been observed to have a prokinetic influence on GI tract muscles [52]. Different types of 5-HT receptor antagonists act through different types of intestine musculature. 5-HT1D antagonists decrease contraction of the circular muscle of human small intestine, whereas 5-HT1B antagonists decrease contraction of the longitudinal muscle [53]. A few publications have distinguished the 5-HT1P receptor as a separate subtype [54,55,56], but it needs to be defined and is suspected to be either the 5-HT7 receptor or a heterodimer consisting of the dopamine D2 receptor with either the 5-HT1B or 5-HT1D receptor with a functional role in the ENS, as it was not detected in the CNS [57]. The 5-HT1P receptor may be critical in the initiation and maintenance of peristaltic and secretory reflexes; it activates submucosal IPANs, and receptor agonists can be expected to enhance diarrhea, whereas antagonists stimulate constipation, or even paralytic ileus [56,57]. Several antidepressant and antimigraine drugs acting via the 5-HT1 receptor have been registered and approved. Sumatriptan prototypical triptane with affinity for different 5-HT1 receptors affects the smooth muscles in the GI tract and, in 1992, was the first triptane approved by the FDA for migraine treatment. Triptanes of mixed affinity to 5-HT1 receptors were available on the market later. Sumatriptan, zolmitriptan, naratriptan, and riazatriptan are full agonists of all 5-HT1 receptors. Frovatriptan is a full agonist to 5-HT1B and 5-HT1D receptors with partial agonism towards 5HT1A and 5-HT1F receptors. Attempts to develop 5-HT1D selective agonists have been unsuccessful, but a new class of drugs was recognized during the research into selective subtypes of 5-HT1 receptor agonists. These are “ditans” that selectively act at the 5-HT1F receptor. Activation of a second messenger for the 5-HT1B receptor, but not the 5-HT1D or 5-HT1F receptors, correspond to the contractile potency of isolated human blood vessels in vitro and in anesthetized canines in vivo [58]. Lasmiditan is considered to be a first-in class “ditan” registered by the FDA in October 2019 [59] and is under evaluation by the EMA. None of the triptanes or lasmiditan have been registered directly for GI tract diseases [60]. However, sumatriptan given to a patient with functional dyspepsia delays gastric emptying, improves gastric accommodation, and reduces the perception of gastric distension and early satiety [61]. Thus, 5-HT1B and 5-HT1D receptors may be involved in the mechanism of vomiting, though this has not been elucidated [62].

The 5-HT2 receptor family consists of three subtypes: 5-HT2A, 5-HT2B, 5-HT2C. This family has excitatory activity, but 5-HT2A also has inhibitory effects. 5-HT2A receptors were first identified in the brain but soon after were found to mediate several effects in the periphery, including platelet aggregation, chloride ion secretion [63], and triggering the contraction of smooth muscles due to its presence in myenteric and submucosal neurons, enterocytes, and both the longitudinal and circular muscles of the GI tract [64]. Many cell types in the periphery express 5-HT2A receptors, including platelets, fibroblasts, lymphocytes, and myocytes [9]. 5-HT2B receptors are present in the longitudinal and circular muscle layers and myenteric neurons. Their activation results in the contraction of smooth muscle in the stomach fundus and longitudinal muscle in the intestine. In preclinical studies, these receptors were postulated to take part in the development of the ENS [5]. 5-HT2C receptors have no known role in GI physiology. There are a number of drugs that act through 5-HT2 receptors. Some are used in migraine, including methysegrid (withdrawn from the market due to cases of retroperitoneal fibrosis), pizotifen, and cyproheptadine. Atypical antipsychotics, such as risperidone, olanzapine, clozapine, and sertindole, block 5-HT2A receptors with high affinity but have limited selectivity versus 5-HT2C and dopamine receptors [8a]. The discovery that 5-HT2A receptor inverse agonists have antipsychotic effects opened an alternative pathway for treating psychosis and drove the development of pimavanserin, a 5HT2A inverse agonist. Pimavanserin lacks dopamine receptor affinity but exhibits antipsychotic activity [65]. In 2016, the FDA and EMA approved pimavanserin for the treatment of psychosis in Parkinson’s disease. Considering the GI system, 5-HT2B receptor expression has been found in spontaneous human carcinoid tumors [66] and hepatitis C-type hepatocellular carcinoma [67]. This suggests 5-HT receptors as a therapeutic target.

There is still little evidence on the expression of 5-HT1C receptor outside the CNS. Low levels of the 5-HT2C receptor mRNA has been noted in pancreatic islet cells [68] and been induced in cultured adipocytes [69]. The physiological role of 5-HT2C in the above-mentioned areas still needs to be elucidated.

The 5-HT3 receptor is different in structure and function from the other six families of 5-HT receptors. The 5-HT3 receptors have a pentameric composition of five identical or nonidentical subunits (from A to E). Only 5-HT3A subunits form functional homomeric 5-HT3 receptors. Other subunits have been identified (from B to E), but only 5-HT3B has been investigated to any extent. Co-expression of the 5-HT3B subunit with 5-HT3A creates the 5-HT3AB receptor. The presence of at least one 5-HT3A subunit is necessary for heteromeric forms [9]. The number and arrangement of 5-HT3C, 5-HT3D, and 5-HT3E subunits in functional receptors has not been determined, though their expression at the protein level was not confirmed too long ago [70]. 5-HT3 receptors create the only group that is ion channel gated by ligand and permeable to sodium, potassium, and calcium. They are expressed mostly in the CNS but are also important in the GI tract, as they take part in gut motility and chemotherapy-induced nausea and emesis. Receptor immunoreactivity was confirmed in the myenteric and submucosal plexuses, cells of Cajal, and the circular and longitudinal muscle cells. Receptor genes were present in both the colon and intestine [71,72]. A preclinical study showed that myenteric IPAN excitation occurs through 5-HT3 receptor activation, and the myenteric and submucosal neurons respond to stimulation [73]. 5-HT3 receptors also play a role in GI mucosal secretion, as their activation results in chloride secretion and serotonin release from EC cells [74,75]. Some studies have confirmed that 5-HT3A receptor mRNA, in particular, is present at high levels in the stomach and colon rather than the small intestine. These receptors are expressed in cholinergic nerves and PDGFRα-positive cells in the myenteric plexus, directly influencing motility. Their activation accelerates gastric emptying and colon transit [76]. 5-HT3 receptors play a role in rotavirus –induced diarrhea. Hagbom showed that serotonin is secreted by EC cells in response to rotavirus enterotoxin NSP4, which acts via 5-HT3 receptor and increases bowel motility [77]. The mechanism of chemotherapy-dependent nausea is via excitation by anticancer drugs (e.g., cisplatin) of EC cells to release serotonin, which activates 5-HT3 receptors and vagal sensory afferent neurons conduct signals to the emetic center in the brain stem [17,78]. In patients treated by cisplatin, higher urinary levels of 5-HIAA confirmed the correlation with the development of emesis. What is important about transmission depends on vagal excitation rather than serotonin release; thus, 5-HT3 antagonist (ondansetron) does not affect the increased 5-HIAA urinary levels [79].

Dependent on anticancer drug doses and type, ondansetron combined with dexamethasone gives complete protection from vomiting in up to 70–80% of cases without adverse effects on intestinal activity and stool consistency [17,80,81]. Ondansetron was first approved in the EU in 1990 and in the US in 1991. It is indicated in the EU for the management of nausea and vomiting induced by cytotoxic chemotherapy and radiotherapy (adults and children > 6 months) and for the prevention and treatment of post-operative nausea and vomiting (adults and children > 1 month). The FDA issued a warning about the risk of abnormal heart rhythms from high doses of ondansetron in 2011, followed by the EMA in 2012 [82]. The SmPC was updated in November 2019 with important changes to the section on “Fertility, pregnancy and lactation”. The SmPC now states that ondansetron should not be used in the first trimester of pregnancy due to the risk of congenital cardiac malformations and oral cleft [83]. Alosetron and cilansetron were implemented into IBS-D treatment but were suspected in the development of ischemic colitis requiring advanced surgical treatment, despite no serious constipation. Due to such serious adverse events, these drugs were withdrawn from the market in 2000. The FDA approved a supplemental new drug application in 2002, which allows for the remarketing of alosetron, but under conditions of restricted use [17,60]. Several other drugs also act on the GI tract via the 5-HT3 receptor. These are non-selective 5-HT3A antagonists, such as metoclopramide, which acts as an antagonist to both dopamine D2 and 5-HT3A receptors. It was approved by the FDA and EMA in 1979 and is recommended to avoid postoperative and chemotherapy-induced emesis. However, in 2013, some restrictions to the use of metoclopramide were introduced by the EMA [84] that were mainly aimed to minimize the known risks of potentially serious adverse neurological reactions. It was recommended that metoclopramide should be prescribed only for short-term use (up to five days) and not be used in children < 1 year of age, and in children > 1 year of age, it may only be used as a second-choice treatment for the prevention of delayed nausea and vomiting after chemotherapy and for the treatment of post-operative nausea and vomiting. In adults, it is used for the prevention and treatment of nausea and vomiting, such as that associated with chemotherapy, radiotherapy, surgery, and in the management of migraine. Moreover, the maximum allowable doses in adults and children were reduced, and higher strength formulations removed from the market [84]. 5-HT3A selective antagonists consist of palonosetron (approved by FDA in 2003 and EMA in 2009 as antiemetic), alosetron (approved and reapproved in 2002 in IBS treatment), granisetron (approved by FDA in 1993 and EMA in 2012 as antiemetic), tropisetron (approved for chemotherapy-induced emesis), and ondansetron (approved by FDA in 1991 as antiemetic in patients receiving chemotherapy and in postoperative emesis) [60].

The 5-HT4 receptor family represents G protein-coupled metabotropic receptors with seven splice variants associated with adenylyl cyclase activity. These receptors are widely distributed within the CNS and periphery, including the heart, GI tract, adrenal and salivary glands, urinary bladder, and lungs [9]. Stimulation of the receptors by an agonist affects the modulation of GI motility and contraction of the esophagus and potentiates peristaltic reflex by colon relaxation and contraction depending on the method of stimulation. In a preclinical investigation, activation of the 5-HT4 receptors located on circular smooth muscle cells resulted in colon relaxation, whereas activation via receptors located on neurons leads to excitatory and inhibitory neurotransmitter release (calcitonin gene-related peptide, vasoactive inhibitory peptide, substance P) [85]. 5-HT4 receptors are important in the early development of enteric neurons, affecting neurogenesis after injury or surgical wound healing after gut anastomosis in adults. Thus, in an animal model, the receptor agonist mosapride promoted the regeneration of an impaired myenteric plexus and recovered the defecation reflex in the rectum [86,87]. In the ileum and right colon, serotonin increases chloride secretion via the 5-HT4 receptors and, with short-chain fatty acids, HCO_3_ luminal secretion [16]. In addition, voltage-controlled ion channel stimulation influences not only intestinal muscles, but also heart atria [88,89,90], which may be responsible for the cardiac toxicity of pure older agonists, such as cisapride. The arrhythmogenic effect of stimulation of atrial 5-HT4 receptors was proposed long ago [91], and the antiarrhythmic effects of 5-HT4 antagonists were presented in a porcine model of atrial fibrillation [92]. Atrial 5-HT4 expression was increased in human chronic atrial fibrillation [93]. QT prolongation and ventricular tachyarrhythmias are thought to be due to potassium channel dysfunction, and cisapride can trigger tachycardia and supraventricular arrhythmia via 5-HT4 atrial receptors [94]. Later, the mechanism through which cisapride promotes cardiac arrhythmias was found to be unrelated to 5-HT4 receptor agonism. The cardiac risk is thought to be related to its affinity for hERG channels, which results in QT prolongation and is enhanced by the concomitant use of inhibitors of CYP involved in cisapride metabolism [95]. Nevertheless, cisapride was withdrawn from the US market in 2000 [76] and suspended from the European market in 2002 [96] due to serious adverse cardiac events. The affinity of the 5-HT4 receptor for particular ligands has been noted to depend on which COOH-terminal splice variant is expressed by a particular cell. This differentiation may explain why the same agonist facilitates increased GI motility with high intrinsic activity, whereas the intrinsic activity in cardiac muscle is low [97,98]. The only drugs acting as selective 5HT4 agonists that remain on the market are mosapride, which improves gastric emptying and reduces gastro-esophageal reflux symptoms, and prucalopride for the treatment of chronic constipation in adults for whom laxatives do not work well enough [99]. Prucalopride has also been investigated in idiopathic, diabetic, and connective tissue disease-related gastroparesis [100,101]. Partial agonists of 5-HT4 receptors, such as tegaserod, seem to be safer by not resulting in heart toxicity and, therefore, are useful in treating the symptoms of IBS-C. Tegaserod reduces the severity of colitis, increases the proliferation of crypt epithelial cells via receptor stimulation, and reduces oxidative stress-induced apoptosis, which was confirmed in animal models before implementation in IBS treatment [102]. Some data also indicate a beneficial role of tegaserod in gastro-esophageal reflux disease (GERD) and gastroparesis treatment. It improves esophageal acid clearance, reduces lower esophageal sphincter (LES) relaxation, and enhances gastric meal passage. This reduces the low postprandial esophagus exposition for stomach acid and decreases GERD symptoms [100,101]. Only a few of the drugs mentioned above were approved in the US and EU with a mechanism acting via the 5-HT4 receptor, though tegaserod was also withdrawn from the market in 2007 due to adverse cardiovascular effects [58].

The 5-HT5 and 5-HT6 receptors are present mostly in the CNS [103,104,105]. They are probably involved in psychiatric disorders, but also anxiety and memory processes [106].

The 5-HT7 receptors have five splice variants that are abundantly present in the vessels and extravascular smooth muscles of the GI tract [5]. They are present in the human stomach, small intestine, and colon, and their activity has been confirmed in myenteric and submucosal IPANs [107]. The activation of these receptors in IPANs produces slow excitatory postsynaptic potentials, resulting in muscle relaxation. In animal models, selective 5-HT7 antagonists have been shown to inhibit the excitatory potential and reduce the accommodation of the circular muscles during the relaxation phase of peristalsis. Overstimulation of the receptors has been postulated to be responsible for abdominal bloating and the exaggerated relaxation of circular muscles, resulting in the symptoms of functional bowel diseases. 5-HT7 agonists may improve after-feeding symptoms, such as postprandial satiety [108]. A similar effect has been investigated for other fundus-relaxing drugs, including sumatriptan and buspirone, which are not classical 5-HT7 agonists. The 5-HT7 receptor is expressed in enteric neurons and CD11c/CD86 cells of the colon in animal models suffering from induced CD. Receptor stimulation has shown an anti-inflammatory effect, likely by the regulation of cytokine production by activated immune cells. The 5-HT7 receptor takes part in cytokine production by antigen-presenting dendritic cells and LPS-stimulated macrophages, except that dendritic cells are able to modulate T-cell function through the 5-HT7 receptor. Thus, the 5-HT7 receptor blockade results in severe inflammation severity, whereas its stimulation results in the clinical remission of symptoms [109]. In contrast to these results, in 2,4,6 trinitrobenzene sulfonic acid-induced colitis, blockade of the receptor does not have such an effect on the severity of inflammation. Thus, this activity may be model-dependent and require further studies [110]. There have been solely antidepressant and antipsychotic drugs approved in the US and EU that have a mechanism of action involving the 5-HT7 receptor [61].

The peripheral effects of acting on serotonin receptors and therapeutic outcomes are presented in Figure 2.

## 3. Conclusions

In summarizing the multimodal role of serotonin, it is worth underlining that the effects of its activity differ according to the receptors on which it acts. Serotonin takes part in both the physiology and pathophysiology of the GI tract. In general, it increases intestinal motility but, depending on the receptors, may play opposite roles in the upper and lower part of the GI tract. In IBS with diarrhea, the plasma level of serotonin is increased, whereas in patients with constipation it is decreased. Thus, the proposed pharmacological treatment involves 5-HT3 receptor antagonism to slow the passage or 5-HT4 receptor agonism as a prokinetic agent. Models of IBD have confirmed that serotonin plays a role in enhancing inflammation, whereas a decreased level of 5-HT is accompanied by a decreased severity of disease signs and symptoms. An anti-inflammatory reaction goes through activation of 5-HT4 and 5-HT7 receptors. Immunology of the response of the 5-HT receptor involves immune cells, including dendritic cells, macrophages, neutrophils, and lymphocytes, and is altered by the pro-inflammatory cytokines TNFα, IL-1β, IL-6, and IFNγ and the anti-inflammatory IL-10. Although some drugs acting on 5-HT receptors are registered in GI tract diseases, there is still much to be elucidated in terms of their mechanism of action, efficacy, and safety.

## Figures and Tables

**Figure 1 molecules-27-01680-f001:**
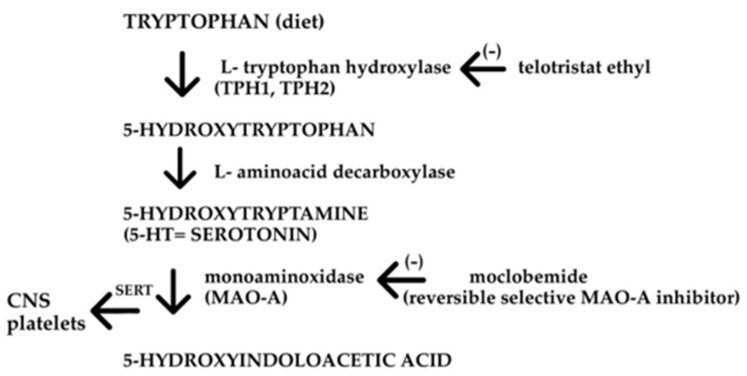
A scheme of serotonin production.

**Figure 2 molecules-27-01680-f002:**
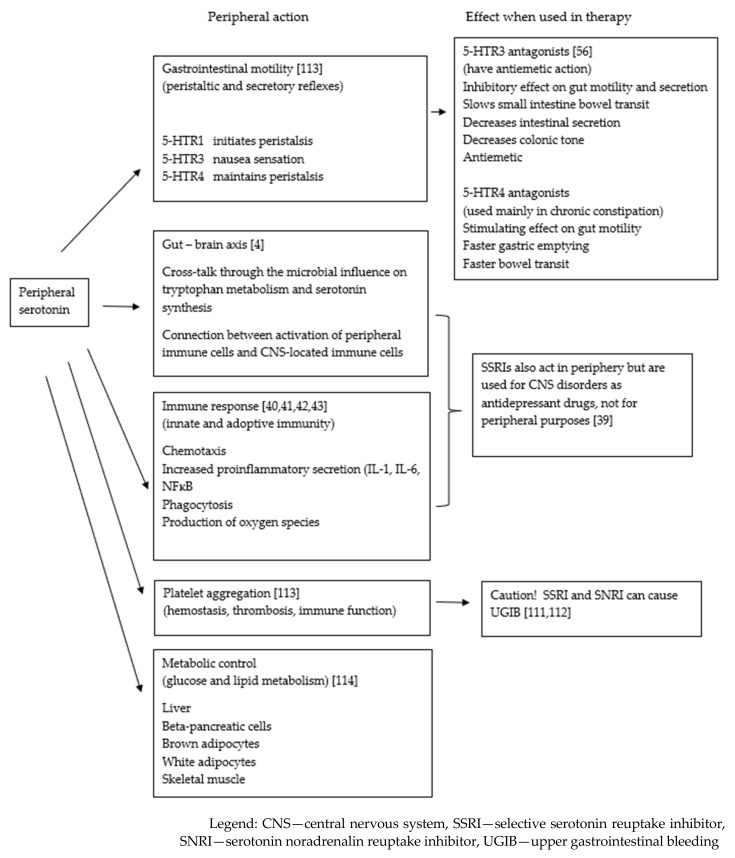
Peripheral effects of acting on serotonin receptors and therapeutic outcomes [111,112,113,114].

**Table 1 molecules-27-01680-t001:** Serotonin receptors and their functions in the gastrointestinal tract.

Receptor Family	Receptor or Subtype	Function
5-HT1	5-HT1A, 5HT1D	Gastric fundus relaxation
	5-HT1B/1D	Prokinetic intestinal stimulation
	5-HT1D	Contraction of intestinal circular muscle
	5-HT1B	Contraction of intestinal longitudinal muscle
	5-HT1P	Peristaltic and secretory reflexes
5-HT2	5-HT2A	Contraction of smooth muscles
	5-HT2B	Contraction of smooth muscles in stomach fundus, relaxation of longitudinal muscle in the intestine
5-HT3	5-HT3	Chloride secretion and serotonin release from EC cells
	5-HT3A	Increase intestinal motility
5-HT4	7 splice variants	Increase intestinal motility, contraction of esophagus, relaxation of colon, chloride secretion
5-HT5	-	Not known in gastrointestinal tract (essential solely in CNS)
5-HT6	-	Not known in gastrointestinal tract (essential solely in CNS)
5-HT7	5 splice variants	Excitatory effect, anti-inflammatory activity

CNS—central nervous system.

## Data Availability

Not applicable.

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
