# Peer review of "The Role of Serotonin Neurotransmission in Gastrointestinal Tract and Pharmacotherapy"

_molecules, 2022, doi:10.3390/molecules27051680_

Round 1

Reviewer 1 Report

This review on “The role of serotonin neurotransmission in gastrointestinal tract and pharmacotherapy ” reporting a short review of specifically on 5-HT neurotransmitter role in GI physiology, it will be useful information for the researcher in the field. Author revised review carefully, can be acceptable in present form.

Reviewer 2 Report

I have no further comments or suggestions.

This manuscript is a resubmission of an earlier submission. The following is a list of the peer review reports and author responses from that submission.

Round 1

Reviewer 1 Report

The article by Guzel and Mirowska-Guzel reviews the role of serotonin and its receptor subtypes in the gastrointestinal tract. It comprises from the synthesis and degradation of this neurotransmitter to its multiple effects in the body, focusing on the effects on the gastrointestinal tract and describes the studies showing the effect of different agonists and antagonists of different serotonin receptors on this system and some other effects.

The article deals with an important subject and provides a good overview of it. However, the English quality is not homogeneous throughout the paper and in general needs to be profoundly revised, especially for pronouns, prepositions, single / plural expression of verbs according to the nouns, style and clarity. The quality of the English does not allow to understand some of the ideas exposed.  Also, some paragraphs need to be separated by full stops, when starting a new subject.

Some conceptual issues that should be revised are:

Lines 41-42 The sentence “5-HT is inactivated by the serotonin reuptake transporter (SERT) or organic cation transporters and dopamine transporter when SERT is inhibited.” is confusing. SERT does not inactivate serotonin, but only relocates it back into cells. Inactivation is produced by its conversion by several enzymes, mentioned in the next sentence. Also, add “by” before “organic cation...”

At the end of the introduction, the authors could include a sentence stating the importance of studying the role of serotonin in the gastrointestinal tract, in view of the health data provided.

Section 2 would benefit from including a scheme showing the various secretion sites, synaptic and paracrine pathways described.

Line 115 change “excretion” to “secretion”

In line 128, the meaning of “share” is not clear.

Line 140: check the phrase “A few publications show that 5-HT1P receptor exists”

Line 150: the sentence “5-HT2 is the family consisted of three subtypes: 5-HT2A, 5-HT2B, 5-HT2C that, except 5-HT2A which also has inhibitory effect, show excitatory type of activity.” Is very confusing. It should be re-written as: “The 5-HT2 receptor family consists of three subtypes: 5-HT2A, 5-HT2B, 5-HT2C that show excitatory activity, with the exception of 5-HT2A, which also has inhibitory effects”

In line 153, please explain what type of influence has 5-HT2AR on smooth muscle contraction.

The dissertation about the approval of Ondansertron and its adverse cardiac effects (lines185-198) lies out of the scope of the review and in my opinion should be removed.Similarly, the dissertation about the authorized uses of metoclopramide should be reduced, focusing on its effects in the gastrointestinal system and its medical uses, and only briefly mentioning the adverse effects and its short-term use authorization.

The two paragraphs about 5-HT5 and 5-HT6 receptors, which do not participate in the gastrointestinal system, could be shortened at a minimum and only mentioned at the beginning of the section to explain that these don´t have effects related to the topic of the review.

It would be interesting to discuss why many antidepressants produce diarrhea at the beginning of the treatment.

The effects on the immune system mentioned in the conclusions should be explained with more emphasis in the main text.

The conclusions can be revised for conciseness and clarity.

Table 1 needs a legend.

Some writing issues are:

Abstract

Line 13-14 I would remove the sentence “It is certainly due to the group of anti-depressive drugs 13 which are selective serotonin reuptake inhibitors (SSRI) widely used all over the world”, since it doesn´t add to the main information necessary in the abstract.

Line 15: add “the” between “In” and “periphery” to read “In the periphery”.

Serotonin synthesis starts by hydroxylation of orally delivered tryptophan, followed by decarboxylation. Serotonin acts through numerous types of receptors. It clinically…   It acts as a paracrine modulator, hormone and growth factor.

The aim of THE present article

Introduction

Line 31-correct the spelling of 5-hydroxytryptamine.

Tryptophan undergoes hydroxylation in neurons and chromaffin cells by the enzyme tryptophan hydroxylase (TPH) that exists…

Line 36: add “a” before “non-specific decarboxylase”. Change “affects” to “participates also in histamine…”

Lint 39: add “the” before “remaining” and before “CNS”.

Line 40: change “EC cells answer for chemical and mechanical stimulation” to “EC cells respond to chemical and mechanical stimulation”.

Line 44: change “secreted in kidneys” to “excreted by the kidneys”

Lines 45-49 could be re-written as: Serotonin was first discovered in the bowel and called “enteramine” by Vittorio Espramer in 1937. Further studies showed that enteramine has the same structure as the serum vasoconstrictor called serotonin described by Rupport in 1947.”

Check the formatting of table 1.

Line 62. Add a full stop after “anti-inflammatory activity” and start a new paragraph with “Digestive diseases…”

Line 78: add a colon between “mucosa” and “which”. Also in line 89, before “which”

The sentence in lines 94-97 is not clear. Please revise the writing.

In line 102-104, the sentences “Microbial transfer between gut with 5-HT reduced and normalized mediated colitis severity. It stands for protective function of microflora with reduced serotonin levels [15]” are not clear. Please revise the writing

Line 167: change “goes” to “occurs”

Line 173: change the order of directly motility to motility directly

Lines 181-185 the sentences need to be re-written with clarity.

In lines 264 and 265, change “gives” to “produces” and “potential” to “potentials”

In line 285 it is not clear what “direction of stimulation” refers to.

Line 287 change “dependently on receptors” to “depending on the receptors it activates, it may…”

Lines 294-295: the sentence “Downregulation of SERT gives opposite effect to inactivation of TRP 294 1 as it results in clinical outcome due to increasing or reducing serotonin level” is not clear. Please revise for clarity.

Author Response

Dear Editor,

thank you very much for your deep, substantive and very helpful review. We introduced nearly all comments as suggested. Due to all required revisions the manuscript has changed essentially and in present form it is difficult indicate precisely places where changes were introduced. We did our best to make the manuscript acceptable for publication.

Conceptual issues:                                                                                                    Lines 41-42 were corrected.                                                                                    The suggested sentence was added at the end of introduction.                              2 figures were added.                                                                                              Line 115, 140, 150 were checked and corrected.                                                    The issue in line 153 was explained.                                                                          Dissertation on ondansetron and metoclopramide was shortened. Some parts were reduced as suggested.                                                                                    5-HT5 and 5-HT6 receptors were only mentioned.                                                The discussion on producing diarrhoea at the beginning of the treatment was added.                                                                                                                      The effect of serotonin on immune system was explained in the main text.            The conclusions were revised, shortened and cleared.                                            Legend to Table 1 was added.

Writing issue:   

Abstract was changed as required, however we think that the sentence about SSRI gives the idea that introducing these drugs to the market attracted attention to serotonin and its action in GI tract.                                                      Introduction was changed in every mentioned point in every mentioned line of the text. Formatting of the table was improved.   

Yours sincerely,

Tomasz Guzel & Dagmara Mirowska-Guzel                                                                                            

Reviewer 2 Report

This review onThe role of serotonin neurotransmission in gastrointestinal tract” reporting a short review of specifically on 5-HT neurotransmitter role in GI physiology, it will be useful information for further investigation. However, there are some grammatical and typographical errors which should be revised before acceptance.

# Page 1, line 31, The word “5-hydroksytryptamine” corrected as “5-hydroxytryptamine”.

# Page 2, Table 1, 5-HT receptor subtypes column need to organised clearly.

 # Page 2, Line 65, the word “ 37,2 million” and “7,9 million” corrected as “ 37.2 million” and “7.9 million”.

Author Response

Dear Reviewer,

thank you very much for your opinion and comment. We have introduced all suggested changes and we hope that in present form the improved version of the manuscripts is acceptable for publication.

Yours sincerely,

Tomasz Guzel & Dagmara Mirowska-Guzel

Reviewer 3 Report

This review tried to summarize the role of serotonin neurotransmission in gastrointestinal tract, including serotonin in GI diseases and pharmacotherapy. The topic is interesting. However, what is new findings of the serotonin neurotransmission in gastrointestinal tract in recent years? What is the mechanism? What is your view related to this topic. It is too general. The most confuse of this review is that the references you referred are too old. I even did not see the near 5 years’ works about serotonin.

Author Response

Dear Reviewer,

thank you very much for your comments. We did our best to add recent findings and publications on serotonin neurotransmission in gastrointestinal tract. We have introduced over 40 new articles, in majority published within last 3-5 years. We hope that these essential changes in the manuscript make it suitable for publication.

Yours sincerely,

Tomasz Guzel & Dagmara Mirowska-Guzel

Reviewer 4 Report

The structure of the article is poor and the topic is not thoroughly discussed. The paper is not strong as a freeform review, and I would suggest rearranging as a systematic review via PRISMA guidelines for the authors to improve the quality and readability of the article. 

Author Response

Dear Reviewer,

thank you very much for your suggestion but on this stage of creating the manuscript the concept does not allow us to make it as systematic review. We have improved significantly the body of the manuscript and hope you reconsider it as "worth publication".

Yours sincerely,

Tomasz Guzel & Dagmara Mirowska-Guzel

Round 2

Reviewer 4 Report

The authors substantially improved the manuscript. It can be considered fit for publication.

Author Response

Dear Reviewer,

Thank you very much for your review.

Yours sincerely,

Tomasz Guzel, Dagmara Mirowska-Guzel